# Targeting Colorectal Cancer Cells with a Functionalised Calix[4]arene Receptor: Biophysical Studies

**DOI:** 10.3390/molecules27020510

**Published:** 2022-01-14

**Authors:** Angela F Danil de Namor, Nawal Al Hakawati, Sami Y Farhat

**Affiliations:** 1Laboratory of Thermochemistry, Department of Chemistry, University of Surrey, Guildford GU2 7XH, Surrey, UK; 2Department of Biological Sciences, Faculty of Science, Beirut Arab University, Tripoli 1107-2809, Lebanon; n.elhakawati@bau.edu.lb; 3Dr. Suliman Habib Hospital, Dubai P.O. Box 500001, SZR, United Arab Emirates; samiyfarhat@yahoo.co.uk

**Keywords:** thioacetamide calix[4]arene, colorectal cancer, ^1^H NMR, Raman

## Abstract

Colorectal cancer (CRC) is a disease which is causing a high degree of mortality around the world. The present study reports the antiproliferative impact of the thioacetamide calix[4]arene, CAII receptor on a highly differentiated Caco-2 cell line. This statement is corroborated by the MTT assay results which revealed a reduction in the cell viability with an IC50 value of 19.02 ± 0.04 µM. Microscopic results indicated that at the starting amount of 10 µM of CAII, a decrease in cells confluency can already be observed in addition to changes in cells morphology. Cell metabolic pathway changes were also investigated. ^1^H NMR findings showed downregulation in lactate, pyruvate, phosphocholine, lipids, and hydroxybutyrate with the upregulation of succinate, indicating a decline in the cells proliferation. Some biochemical alterations in the cells as a result of the CAII treatment were found by Raman spectroscopy.

## 1. Introduction

Colorectal cancer (CRC), also known as bowel cancer, affects over 1.93 million people worldwide and has led to almost 935,000 deaths according to the report published by the World Health Organization (WHO) in March 2021. It is the third most commonly detected malignancy and the fourth global killer among cancers [1]. As far as Europe is concerned, CRC is the most common cause of death after lung cancer [2], while in the United Kingdom this disease occupies the third position after breast and lung cancer, as recently reported by the National Health Service [3]. As the world population increases, CRC incidence is expected to increase by 60%, with more than 2.2 million new cases and 1.1 million deaths by 2030 [4]. However, while improvements in perioperative care along with chemotherapy or radiotherapy will help in decreasing the CRC mortality [5], the scope of these treatments is limited due to (i) their non-selective nature since they target not only cancer cells, but also healthy ones, and (ii) other side effects encountered during and after treatment [6,7,8,9]. The major drawbacks in these therapies have led to extensive studies in the last three decades aimed at designing selective anticancer agents targeting only cancer cells such as those based on nanotechnology, surface modifications of polymer drugs, and the use of cancer biomarkers [7]. Therefore, it is of outmost importance to address the issue of selectivity in the design of anticancer agents. Given that this is one of the main features of supramolecular chemistry, receptors such as cyclodextrins, crown ethers, calixarenes, and their derivatives have shown a great potential for biomedical applications. Calixarenes have received particular attention in the pharmaceutical field as active anticancer, antidiabetic, anti-obesity [8], and antibacterial agents [9]. Thus, studies on their anticancer properties have been the subject of several research groups [10,11], where investigations in vitro against different types of cancer cell lines were conducted. Baggetto and co-workers [12] have synthesised a number of calix[4]arene-based compounds (calix[4]arene dihydroxyphosphonic acid, *p*-*tert*-butyl calix[4]arene dihydroxyphosphonic acid, and *para*-octanoyl calix[4]arene dihydroxyphosphonic acid) against different brain, breast, liver, blood, and skin cancer cells (MU2, MU2F, HT 1080, SP6.5, IPC227, Jurkat, MEWO, H1-60, Huh7, Hep-G2). The anticancer activity of these macrocycles has been compared to standard anticancer drugs. The outcome of their research revealed that these calix[4]arene derivatives have a potent anticancer activity, especially in melanoma cells (skin cancer) and lymphoblastic leukaemia cells. Similarly, other research groups have tested a calix[4]arene functionalised with four platinum (II) centres on non-small cell lung, breast, and hepatocellular cancer, showing that these compounds offer a better activity than carboplatin (chemotherapeutic drug) [13]. Several reports on a wide range of calixarene derivatives highlighted their remarkably low degree of toxicity [14,15,16,17], to the extent that studies on *p*-sulphonatocalix[4]arene have shown the compound to have zero toxicity toward normal healthy blood cells in concentration levels up to 5 mmol dm^−3^ in vitro and in vivo at doses up to 100 mg kg^−1^ [18]. Few studies have been conducted in vivo on the applications of calix[4]arene derivatives in mammalian models (cancer mice models). Hulίková and co-workers [19] examined the effect of glycol-conjugates of calix[4]arene in a B16F10 mouse melanoma model, where a significant reduction in tumour growth was observed within two weeks. As for clinical investigations, a US clinical trials database recorded one test started in 2012 using a calix[4]arene-based compound as angiogenesis antagonist with no reported results [9].

Metabolomics approaches have been recently developed as complementary technologies to genomic and proteomic fields [20,21], enabling the identification of products from biochemical reactions, and therefore information about metabolic pathways within the cell [22] can be obtained. Moreover, cell metabolites and biomolecules can be identified, thus providing a full metabolic fingerprint of the tested living cell. Despite the above-mentioned research efforts, to our knowledge there is no work reported in the literature regarding the use of functionalised calixarene receptors in colon carcinoma Caco-2 cells (colorectal cancer). Therefore, the aim of this investigation is to assess the bioactive impact of a calix[4]arene derivative, namely 5,11,17,23-tetra-*tert*-butyl[25,27-bis(diethylthiocarbamoyl)oxy]calix[4]arene (CAII) (Figure 1), on this type of cancer cell line using a variety of techniques.

## 2. Materials and Methods

### 2.1. Materials

The cell line used throughout the study of *Caco-2* (human colorectal adenocarcinoma) was obtained from the European Collection of Cell Cultures (ECACC), Porton Down, UK (Culture Collections, 2013). The culture medium and reagents Dulbecco’s Modified Eagle Medium (DMEM); heat inactivated foetal bovine serum (FBS); MEM non-essential amino acids solution (NEAA) (10 mM), 0.5% trypsin-EDTA solution, L-glutamine solution (0.2 × 10^−3^ mol dm^−3^), 10,000 U/mL penicillin/10,000 μg/mL streptomycin (antibiotic); and Dulbecco’s phosphate-buffered saline (DPBS) (1×) Ca^2+^/Mg^2+^ free were purchased from Invitrogen Ltd. (Paisley, UK). 5-Diphenyltetrazolium bromide (MTT) and dimethyl sulfoxide (DMSO) were obtained from Sigma-Aldrich, Ltd. (Poole, UK). Staurosporine from *Streptomyces* sp. and an Annexin V-FITC Apoptosis Detection Kit were purchased from Abcam and deuterium oxide (D_2_O) (D, 99.9%) from Cambridge Isotope Laboratories, Inc, Tewksbury, Massachusetts, United States.

### 2.2. Synthesis of Thioacetamide Calix[4]arene Derivative, 5,11,17,23-Tetra-Tert-Butyl[25,27-bis(Diethylthiocarbamoyl)oxy]calix[4]arene, CAII

The receptor was previously prepared by Danil de Namor and Pawlowski [23] and the synthetic procedure used for its preparation is described in the Supplementary Information.

### 2.3. Cell Culture

Caco-2 cells were cultured in Dulbecco’s Modified Eagle’s medium (DMEM), which is high in glucose (25 mmol dm^−3^), supplemented with 4 mmol dm^−3^ L-glutamine, 1% penicillin/streptomycin, 10% foetal bovine serum (FBS), and 1% non-essential amino acids (NEAA). The cells were maintained at 37 °C in a culture incubator in a 5% CO_2_/95% air atmosphere. They were sub-cultured to approximately 80–90% confluence and afterwards they underwent passaging.

### 2.4. CAII Treatment

Caco-2 cells were seeded in 96-well plates with a density of 1 × 10^4^ cells/well in 0.2 cm^3^/well of a culture medium. Following overnight incubation to allow cell attachment, the culture medium was discarded and treatment with different concentrations of CAII (1, 5, 10, 20, 50, 70, and 100 µM) was applied with a total volume of 0.2 cm^3^/well of serum-free medium incubated for 24 or 48 h. Then, cells were harvested and submitted for cell toxicity determination flow cytometry and ^1^H NMR analyses.

### 2.5. CAII Preparation

The compound was prepared in a minimum volume of dimethyl sulfoxide which did not exceed 0.1% (*v*/*v*), after which dilutions were made in DMEM according to the investigated concentrations. Cultures were checked microscopically (Zeiss TELAVAL inverted light microscope) during the treatment for changes in the cells’ morphology.

### 2.6. Cell Viability

The count of viable cells was determined by the MTT (3-(4, 5-dimethylthiazolyl-2)-2, 5-diphenyltetrazolium bromide) assay. The first step of the procedure was the aspiration of the spent medium following the treatment period. Fresh, serum-free medium containing MTT (0.5 mg/cm^3^, 0.2 cm^3^/well) was added to each well and incubated at 37 °C for 150 min. Following the incubation period, the medium containing MTT was discarded and the obtained formazan precipitate was dissolved in DMSO (0.2 cm^3^). The absorbance was measured at 570 nm using a SPECTROstar absorbance 96-well microplate reader (Omega Software, BMG LABTEACH, Aylesbury, UK) (Lambda 25, Perkin Elmer, Cambridgeshire, UK). The cell viability (percentage) of three or more dependent and independent experiments was calculated using the following equation (Equation (1)):Cell viability (%) = (OD of treated cells)/(OD of control cells) × 100%(1)
where OD is the notation used to indicate the optical density of the culture.

### 2.7. Apoptosis Analysis by Annexin V-FITC Staining

Caco-2 cells (1 × 10^6^ cells/5 cm^3^/flask) were seeded in 25 cm^2^ flasks followed by the treatment of CAII (10 μM). When cells reached 50% confluence, they were washed with PBS (2 cm^3^) after which they were incubated for 24 h at 37 °C in a humidified atmosphere with CO_2_ (5%). Cell pellets were then re-suspended in a binding buffer (500 μL, (1:10)) mixed with Annexin V-FITC and propidium iodide and incubated in a dark area at room temperature (22 ± 2 °C) for 5 to 10 min. The samples were kept in ice until they were analysed using an Applied Biosystems™ Attune NxT Flow Cytometer (Thermo Fisher Scientific, Waltham, MA, USA). The quantitative measurement of cells undergoing death was determined. Forward and side scatters were used for gating to exclude cell debris. Fluorescence emission was detected in the FL-1 channel (519 nm) for cells labelled with Annexin V-FITC and in the FL-2 channel (617 nm) for cells labelled with PI. For each experimental sample, a total of 10,000 events were acquired for Annexin V-FITC and PI.

### 2.8. Structural Insights on the CAII and Caco-2 Interactions Using ^1^H NMR Measurements

Samples (Caco-2 cells and Caco-2 cells treated with CAII) were cultured as described above. The cells were exposed to the receptor (10 µM) for a period of 24 h. Then, Caco-2 cells were collected and suspended in 1× cold phosphate-buffered saline. The samples were rinsed twice with PBS and PBS mixed with deuterium oxide (NMR solvent) to minimise the NMR signal interference with water. Samples were transferred into 5 mm NMR tubes using DSS (4,4-dimethyl-4-silapentane-1-sulfonic acid) internal standard (δ_H_ = 0.00 ppm). All ^1^H NMR spectra were recorded at 298 K on a Bruker DRX-500 pulse Fourier transform NMR spectrometer. Following the ^1^H NMR runs, spectra were collected and the signals of the cell chemical groups were assigned to identify its metabolic profile. Moreover, the interaction mechanism with CAII was determined from the obtained chemical shifts.

### 2.9. ^1^H NMR Spectroscopy of Caco-2 Cell Extracts

^1^H NMR studies were carried out at the University of Surrey to identify the different metabolites in the Caco-2 samples. These measurements were taken at 298 K. The operating conditions included a pulse or flip angle of 30°, spectra width (SW) of 15 ppm, spectral frequency (SF) of 500.150 MHz, delay time of 0.3 s, acquisition time (AQ) of 3.17 s, and line broadening of 0.3 Hz. All ^1^H NMR spectra were processed and base-line corrected using ACD/NMR Processor 12.

### 2.10. Confocal Raman Microscopy Studies

Caco-2 cells were cultured and seeded with CAII using the procedure described in Section 2.3. However, only a 24 h incubation period was used in this study. Cells were fixed in paraformaldehyde (4%), after which they were submerged in PBS (phosphate-buffered saline) prior to analysis. The samples were analysed using an inVia^TM^ Confocal Raman Microscope (Renishaw, New Mills, UK) with a Nd:YAG laser (532 nm, frequency-doubled laser) with an output power of 40 mW and excitation spot of around 1 µm. A GaAlAs laser (wavelength 782 nm) was used for excitation, producing a maximum of 9 mW at the focal plane of a 50× objective, typically used to illuminate the sample. Measurements were carried out at a spectral resolution of 4 cm^−1^.

### 2.11. Statistical Analysis

Data are presented as the mean ± SD. Analyses were conducted using a one-way analysis of variance (ANOVA) and *p* < 0.05 was considered to be statistically significant.

## 3. Results and Discussion

### 3.1. CAII Cytotoxicity against Caco-2 Cell Viability

The dose effect of CAII on Caco-2 cells viability was examined colorimetrically using 3-(4,5-dimethylthiazol-2-yl)-2,5-diphenyltetrazolium bromide (MTT) assay. Cells were treated with different concentrations of the receptor (1 to 100 µM) for 24 h, after which the minimal inhibitory concentration IC_50_ (the concentration of the drug that reduces 50% of the cell viability) was determined. Analysis of the dose response viability inset (Figure 2) shows a significant decrease in the percentage of cell proliferation from 98% at 5 µM to 9% at 50 µM of CAII compared to control cells; therefore, an extended period of incubation was not required (48 or 72 h) since more than 90% of the cells’ growth was inhibited after 24 h. The IC_50_ value for these cells was 19.02 ± 0.04 µM, suggesting that the cytotoxic effect of the tested calix[4]arene derivative was prominent against the Caco-2 cell line.

### 3.2. CAII Inducing Death on Caco-2 Cells

The outcome of the cytotoxicity assay provides strong evidence of the intensive effect of the CAII receptor on Caco-2 cells at high concentrations (~25–50 µM) after 24 h of exposure. As far as the effect of the receptor at lower concentrations on Caco-2 cells is concerned, this was further investigated by using cell death assay. Inducing apoptosis in cancer cells has been recognized as the most recommended strategy in the application of anticancer agents [24].

The results displayed in Figure 3 show the percentage of apoptotic cells (48.3%) treated with CAII (10 µM) for 24 h vs. 6.68% for untreated Caco-2 cells, reflecting the effect of CAII on inducing apoptosis in the investigated cancer cells. Many studies have demonstrated that apoptosis is the best way to get rid of pre-cancerous or cancerous cells [25].

### 3.3. Morphological Changes of Caco-2 Cells Induced by CAII Treatment

Morphological and density changes of Caco-2 cells in response to CAII treatment during the 24 h incubation period were investigated using phase-contrast and bright field microscopy; micrographs are shown in Figure 4 and Figure 5. As shown in Figure 4A, the control Caco-2 live cells appear to be more confluent. However, this confluency decreases with increasing concentrations of CAII. Interestingly, in addition to a lower cell density, morphological changes of the cells were observed with CAII treatment (10 µM) (Figure 5B) when compared with untreated cells (Figure 5A).

For viewing the cells in 3D cultures, laser confocal scanning microscopy was applied, as it has been recognised as a well-established approach with a high resolution, sensitivity, and a penetration depth of up to several millimetres [26]. Figure 6 shows the 3D topographical images of the Caco-2 and receptor-treated cells. Inspection of the micrograph (Figure 6a) reveals the presence of well differentiated cells with a clear cell membrane (red contour) and inner cellular compartments (blue regions). Dramatic changes in the cell morphology were detected following the treatment with CAII (Figure 6b), where disruptions in the cell boundary (membrane) in addition to changes in the redistribution of the intracellular compartments (blue region) were observed, which could be due to the shedding or loss of the cellular material [27].

This finding triggers a question about the mechanism of action of CAII on the metabolic pathways of the cells. One-dimensional ^1^H NMR analysis was conducted to determine the metabolic fingerprint of the treated and untreated cancer cells.

### 3.4. Metabolite Analysis of Untreated and CAII-Treated Caco-2 Cells Using ^1^H NMR Spectroscopy

The proton NMR approach has been used in several studies due to its capability of detecting biomolecular compounds with a high reproducibility [28,29,30,31]. ^1^H NMR analyses of differentiated cells of Caco-2 and Caco-2 treated with CAII were conducted. Several biomolecules were found in the obtained spectra. Metabolic status along with some potential metabolic markers in the samples was identified using the Human Metabolome Database, version 3.0 [32]. Initially, signals for the most significant metabolites, presented in the region of δ 5.00–0.80 ppm, were observed (Figure 7 and Table 1), including glucose, amino acids, phospholipids, tricarboxylic acid intermediates, and other metabolites. Some metabolites were not fully identified due to their overlap with other signals. Overall, thirty-two metabolites were identified as Caco-2 cellular constituents.

A further ^1^H NMR run of Caco-2 cells treated with CAII was conducted (Figure 7) where metabolic changes were screened. A slight upfield shift was observed, suggesting a possible interaction of the cell metabolites with the calix[4]arene receptor.

Screening the metabolic changes as a result of CAII treatment showed reduced levels of glucose as well as water soluble metabolites such as lactate, acetate, hydroxybutyrate, and lysine as compared to the untreated cells. These changes suggest a shift in the metabolic pathway from homolactic to mixed acid formation (lactate, acetate). The outcome of this study revealed some changes in the metabolic profile of the Caco-2 cancer cell (Figure 8) resulting from the CAII treatment. These findings may add further biochemical data at the molecular level in colorectal cancer research, which could be useful in the exploration of cancer treatment.

### 3.5. Confocal Raman Spectroscopy Measurements

Raman Spectroscopy measurements were performed on untreated and CAII-treated Caco-2 cells in order to monitor the effect of the applied calix[4]arene derivative on components of the cells. Figure 9 displays the Raman results of the untreated cancer cells versus the treated ones showing the overall cell response to the applied calix[4]arene receptor. Assignments along with the intensities of the Raman peaks that correspond to some regions that have biological significance are presented in Table 2. The regions of interest displaying the biochemical fingerprint composed of proteins, lipids, and nucleic acid backbone signals are shown in the 500 to 3000 cm^−1^ range. As shown in the spectra of both samples, the band attributed to the phospholipid components of the cell membrane in Caco-2 cells at 1338 cm^−1^ is characterised by a lower intensity than that observed in the Raman spectrum of the CAII-treated cells (Table 2). Bands vibrating in 3055–2882 cm^−1^ appeared to be different in both spectra. Moreover, the blue shift observed in the protein peaks is an indication of a change in the characteristics of this biological material. Changes in the peak attributed to the CH_2_ of the lipid component of the membrane at 1440 cm^−1^ were observed as a result of Caco-2 incubation with the CAII receptor.

## 4. Conclusions

From the above discussion, the following was concluded:

(i)CAII inhibits the Caco-2 cell proliferation in a concentration-dependent manner. The compound mode of action is mediated by an apoptosis mechanism that is considered to be an important aspect in defeating cancer.(ii)The MTT assay results demonstrate the anticancer effect of CAII against Caco-2 cells with an IC50 value of 19.02 ± 0.04 µM, revealing the fast action mechanism of this receptor relative to other reported chemotherapeutic drugs.(iii)Flow cytometry analysis shows that, at a starting dose of 10 µM, an apoptotic cascade in Caco-2 cells is observed. This cell death mechanism was corroborated by ^1^H NMR and Raman spectroscopic analyses. Thus, the reduction of the level of water soluble metabolites treated with CAII as well as the phospholipid component of the cell membrane clearly indicate a change in the metabolic profile of the cancer cell following the treatment with the calix[4]-based receptor.(iv)Raman studies provide evidence that the calix[4]arene–thioacetamide receptor interacts with the protein metabolites of Caco-2 cells. This is indeed an important aspect to highlight in this paper since it proves the effect of the applied anti-cancer agent on the protein transporters which play an essential role in the efflux mechanism of drugs, thus leaving the cells resistant to them.

## Figures and Tables

**Figure 1 molecules-27-00510-f001:**
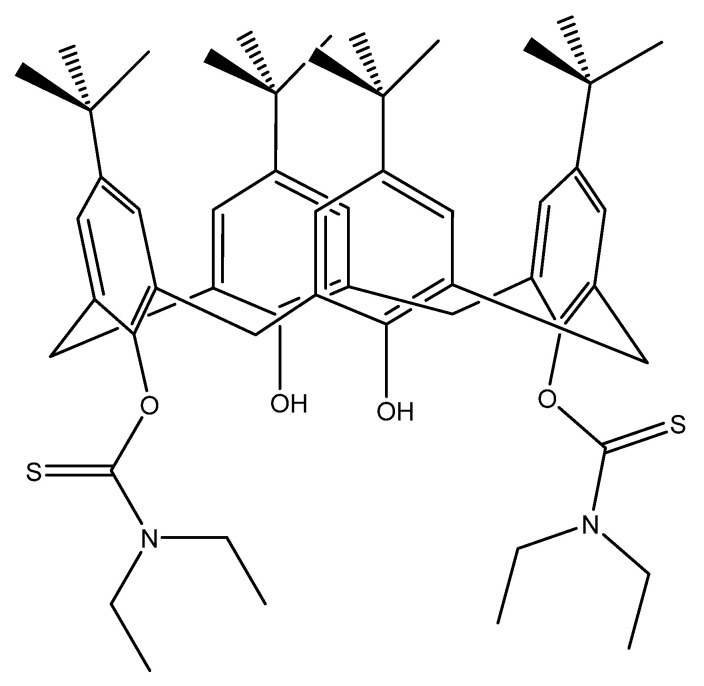
2D structure of 5,11,17,23-tetra-*tert*-butyl[25,27-bis(diethylthiocarbamoyl)oxy]calix[4]arene, CAII.

**Figure 2 molecules-27-00510-f002:**
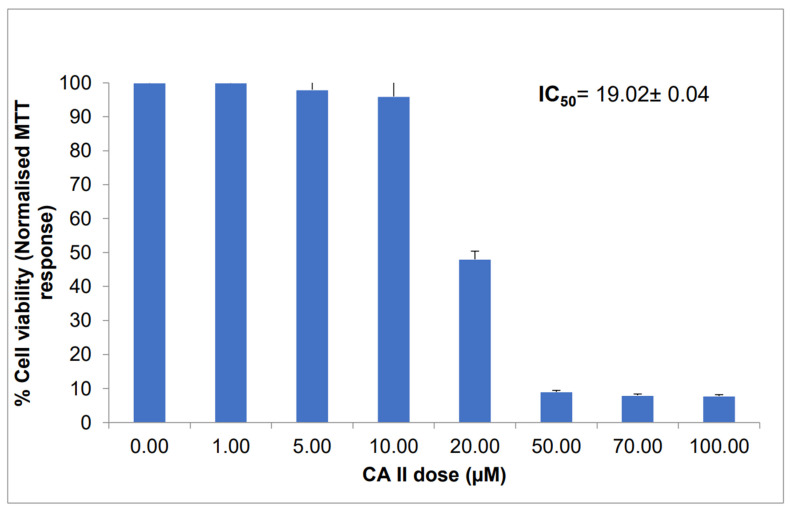
Dose effect of CAII on colon cancer cell viability (Caco-2) after 24 h incubation. Data are shown as mean ± SEM (*n* = 3 dependent and independent experiments). Data analysis was performed using one-way ANOVA followed by Bonferroni post-test, *p* < 0.05 versus CAII.

**Figure 3 molecules-27-00510-f003:**
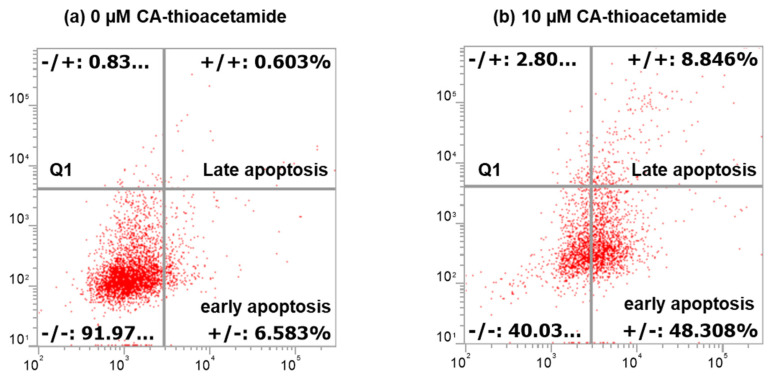
Histogram depicts apoptosis induction with CAII on Caco-2 using flow cytometry technique after 24 h incubation. Scatter plots (**a**,**b**) represent (**a**) untreated cells with 0 μM CAII and (**b**) cells treated with 10 μM CAII. Cells were stained with propidium iodide (PI) and Annexin V-FITC (AV). Live cells = are living unaffected cells; early apoptotic = cells binding Annexin V; late apoptotic = cells binding Annexin V and PI; Q1 indicates necrotic cells due to mechanical damage binding propidium iodide.

**Figure 4 molecules-27-00510-f004:**
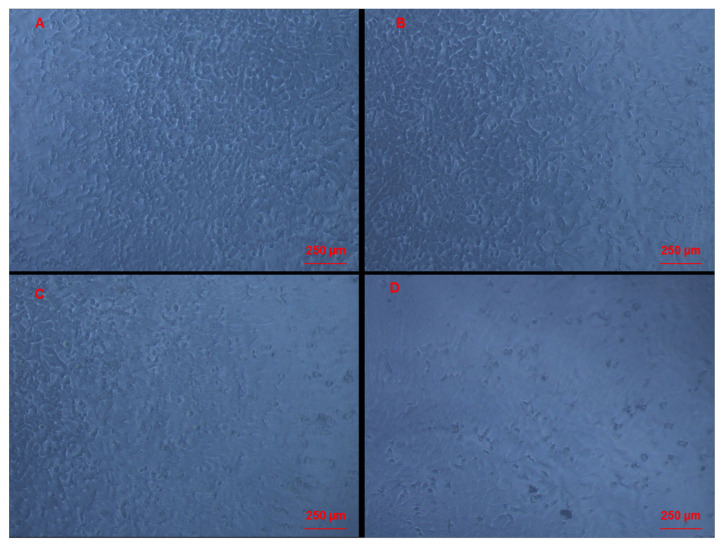
Caco-2 cells observed under the inverted phase-contrast microscope 24 h after being treated with different concentrations of CAII: (**A**) control cells, 0 µM CAII; (**B**) cells treated with 10 µM of CAII; (**C**) cells treated with 20 µM CAII; and (**D**) cells treated with 50 µM CAII. Magnification ×10.

**Figure 5 molecules-27-00510-f005:**
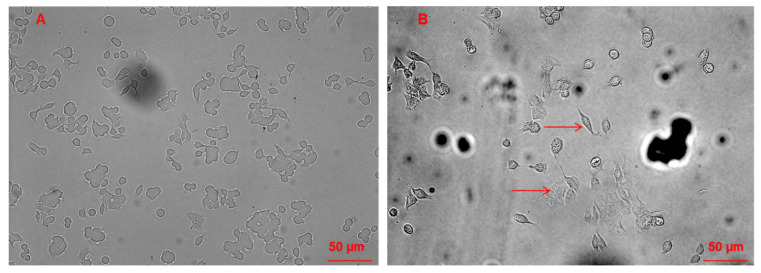
Bright-field micrographs of (**A**) Caco-2 cells and (**B**) Caco-2 cells incubated with 10 µM CAII for 24 h.

**Figure 6 molecules-27-00510-f006:**
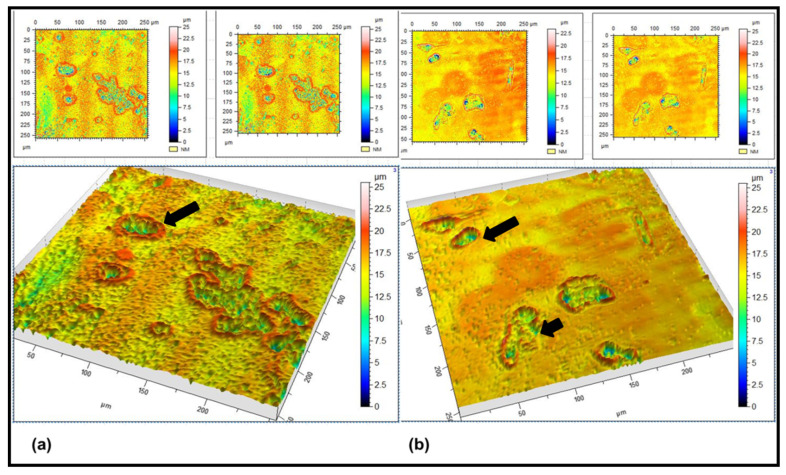
3D Laser scanning confocal microscope topography images of (**a**) Caco-2 cells and (**b**) CAII-treated Caco-2 cells. (**a**,**b**) A view from the top. Cell inner compartments are given in blue. The 3D shape of the cells is shown by red contour lines.

**Figure 7 molecules-27-00510-f007:**
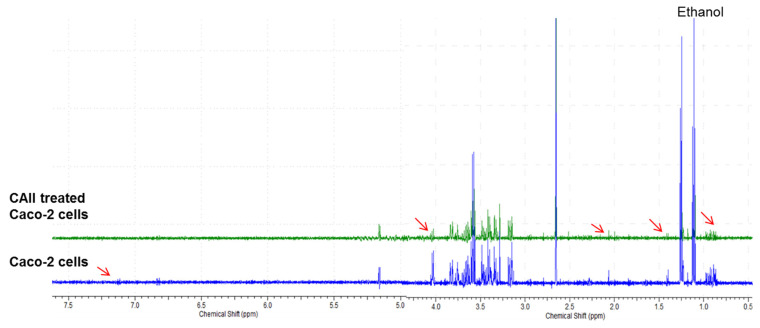
^1^H NMR of differentiated Caco-2 cells and CAII-treated Caco-2 cells in D_2_O showing extracellular metabolites. Number of scans was 132. Region between 5 and 4.20 ppm was omitted due to water suppression.

**Figure 8 molecules-27-00510-f008:**
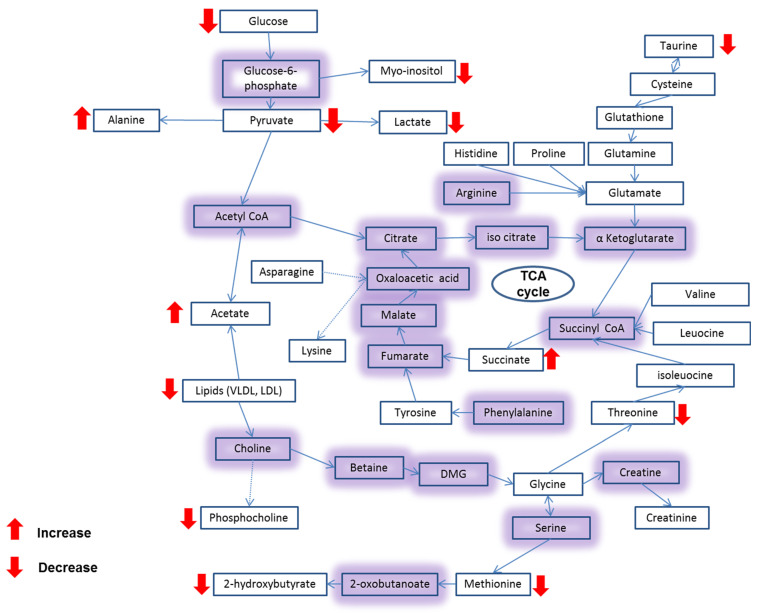
Schematic overview of the metabolic changes in CAII-treated Caco-2 cells.

**Figure 9 molecules-27-00510-f009:**
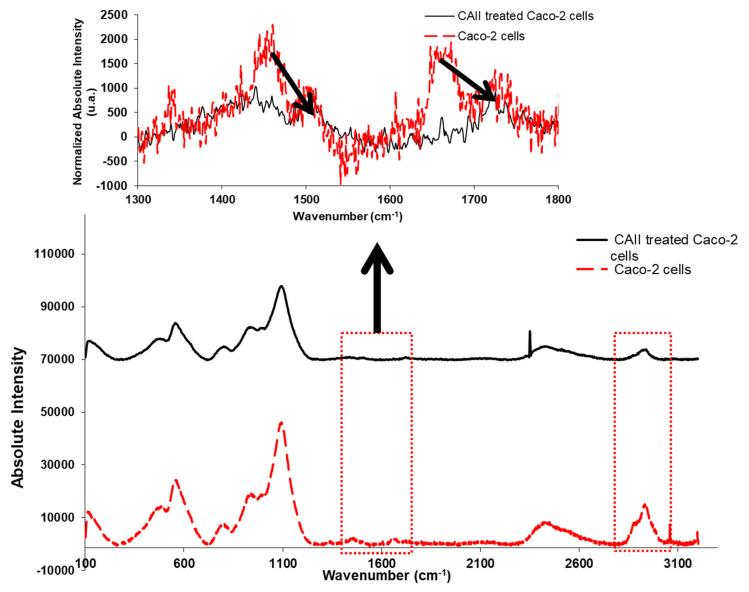
Raman spectra (286–3201 cm^−1^) of Caco-2 and 10 µM CAII-treated Caco-2 cells incubated for 24 h.

**Table 1 molecules-27-00510-t001:** ^1^H NMR assignments of the major identified metabolites in Caco-2 and CAII-treated Caco-2 cells.

Metabolite Number	Metabolites	Caco-2 Chemical Shifts (ppm)	>Caco-2 Treated with CAII Chemical Shifts (ppm)
1	Lactate	1.33, 4.12	-
2	Acetate	1.91	1.91
3	Glutamine	2.12, 2.16, 2.45	2.12,2.16
4	Glutamate	2.05, 2.36	2.05, 2.36
5	VLDL	0.88, 1.29	0.88, 1.29
6	LDL	0.84	-
7	Glycoprotein	-	2.10
8	Lysine	1.44, 1.75, 3.02, 3.76	1.44, 1.75, 3.76
9	Asparagine	2.94, 3.95	2.94, 3.95
10	Creatinine	3.04, 3.93	-
11	Methionine	2.14	-
12	α-Glucose	3.45, 3.54, 3.71, 3.73, 3.85	3.45, 3.54, 3.71, 3.73, 3.85
13	β-Glucose	3.23, 3.43, 3.49, 3.90	3.43, 3.49, 3.90
14	Glycine	3.56	3.56
15	Myo-Inositol	3.55, 3.63, 4.07	3.55, 3.63, 4.07
16	Taurine	3.27	-
17	Glycerophosphocholine	3.24	-
18	Glutathione	2.57, 2.97	2.57, 2.97
19	Threonine	1.34, 4.27	1.34
20	Succinate	2.39	-
21	Alanine	1.49	1.49
22	Tyrosine	6.91	-
23	2-hydroxybutyrate	1.19, 2.28	2.28
24	Citrate	2.70	2.70
25	Pyruvate	2.37	2.37
26	Histidine	7.11	-
27	S-Sulfocysteine	3.48, 3.66	3.48, 3.66
28	O-phosphocholine	4.17	4.17
29	Carnitine	3.17	3.17
30	Valine	1.05	1.05
31	Isoleucine, leucine	0.94, 0.97	0.94, 0.97
32	Ethanolamine	3.16, 3.82	3.16, 3.82
33	AMP	4.01, 4.15, 4.36, 6.15	4.15, 6.15

**Table 2 molecules-27-00510-t002:** Intensity values at some regions of interests of Raman spectra for Caco-2 cells and CAII-treated Caco-2.

Assignments	Wavenumber (cm^−1^)	Intensity Caco-2 Cells	Intensity CAII-Treated Caco-2 Cells
C=C (aromatic)	3055	885	85
CH_3_ (stretching)	2971	6720	1204
CH_2_ (stretching)	2882	8309	1480
C=O (stretching)	1728	913	591
Amide I	1650	1162	−28
CH_2_ deformation of nucleic acids	1450	1576	583
CH_2_ twist and bend	1338	625	124
PO^2−^ symmetric stretching	1108	41,436	25,425
C–O (ribose)	982	17,719	11,463
Skeletal mode of polysaccharides	939	18,311	12,195
Cytosine and uracil (stretching)	789	6985	4398
>S-S< (stretching)	550	23,519	13,055
>S-S< (stretching)	511	12,230	7169

## Data Availability

Not applicable.

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
