# Peer review of "Targeting Colorectal Cancer Cells with a Functionalised Calix[4]arene Receptor: Biophysical Studies"

_molecules, 2022, doi:10.3390/molecules27020510_

Round 1
Reviewer 1 Report
This manuscript has several major concerns such as:
- 4. Metabolite analysis of Caco-2 and CAII treated Caco-2 cells using 1H NMR spectroscopy. For all NMR specters signals are at the noise level, and it is difficult to assess chemical shifts and assign structures.
.
- Raman spectra decoding:
«As shown in the spectra of both samples, the band attributed to the phospholipid components of the cell membrane in Caco-2 cells is characterised by a lower intensity than that observed in the Raman spectrum of the CAII treated cells» - What lane is this? where did it appear? What is the value in cm-1? Where is in the table?
«Moreover, the blue shift observed in the protein peaks is an indication of a change in the characteristics of this biological material» The blue shift in spectra is not detectable
The normalized spectrum from 1300-1800 cm-1 is too noisy to be asserted in the shifts drawn by the arrow.
- The solubility data on (CAII) should be specified. Is it a buffer solution or organic solution?
- Please, specified the solubility of CAII in cell culture?
Additionally, the toxicology of studied calixarene should be studied, from point of view, effect to healthy cells?
Reviewer 2 Report
My Comments were:
In this manuscript, Angela F Danil de Namor et al. studied anticancer
effect of thioacetamide calix[4]arene. It is very interesting and
organized well. Basically, I think it could be accepted after
considering these points as follows.
1. The authors should give information about using solvents in all
experiments ( in experimantal section, in captions of figure). I have
doubts about solubility of the calixarene in water and close to it solvents.
2. The authors should add information about toxicity of calixarene and
used solvents on healthy cells .
3. The authors should check the references - different formalizations.
And question about reference on report in conference. Maybe this speaker
has the article?
But I do not see toxicity of using calixarene and solvent on the healthy cells. In introduction are presented information of toxicity about derivatives of calixarene but not using structure. Ethanol and methanol as alcohols have different toxicity. Information about toxicity of solvent and structure of using molecules is important if we discuss about receptors for Colorectal cancer.
Reviewer 3 Report
The authors partially answered of my questions asked in the first version of the manuscript (molecules-1410514). But knowledge of toxicity of novel compounds with anticancer activity is required. The study does not assess the impact of test compounds on normal cells. The anticancer activity data is not relevant without this information.
Round 2
Reviewer 3 Report
It is well-known that sulfonated derivatives of macrocyclic compounds have low toxicity. However completely different compound, namely disubstituted p-tert-butilthiacalix[4]arene, is studied in the manuscript molecules-1489411. The toxicity of this compound is unlikely to be low.
So, the authors did not add information about the toxicity of the studied compound on normal (or healthy) cells. Unfortunately, I can’t recommend this manuscript for publication.
